# Adsorption Equilibrium and Mechanism and of Water Molecule on the Surfaces of Molybdenite (MoS2) Based on Kinetic Monte-Carlo Method

**DOI:** 10.3390/molecules27248710

**Published:** 2022-12-08

**Authors:** Ruilin Wang, Xinyu Wang, Zhijun Zuo, Shijun Ni, Jie Dai, Dewei Wang

**Affiliations:** 1Faculty of Geosciences and Environmental Engineering, Southwest Jiaotong University, Chengdu 611756, China; 2Department of Geochemistry, Chengdu University of Technology, Chengdu 610059, China; 3Key Laboratory of Sedimentary Basin and Oil and Gas Resources, China Geological Survey, Ministry of Land and Resources & Chengdu Center of Geological Survey, Chengdu 610081, China; 4Key Laboratory of Coal Science and Technology of Ministry of Education and Shanxi Province, Taiyuan University of Technology, Taiyuan 030024, China

**Keywords:** molybdenite oxidation, weathering, adsorption mechanism, annealing dynamics, kinetic Monte Carlo modeling

## Abstract

The oxidation/weathering of molybdenite (MoS_2_) is too slow to be monitored, even under pure oxygen and high temperatures, while it proceeds rapidly through humid air. The adsorption of water molecules on molybdenite is necessary for the wet oxidation/weathering of molybdenite. Therefore, we employ kinetic Monte Carlo modeling to clarify the adsorption isotherm, site preferences and kinetics of water on different surfaces of molybdenite. Our results indicate that (1) the adsorption capacity and adsorption rate coefficient of H_2_O on the (110) surface are significantly larger than those on the (001) surface at a temperature of 0~100 °C and a relative humidity of 0~100%, suggesting that the (110) surface is the predominant surface controlling the reactivity and solubility of molybdenite in its interaction with water; (2) the kinetic Monte Carlo modeling considering the adsorption/desorption rate of H_2_O, dissociation/formation rate of H_2_O and adsorption/desorption of dissociated H indicates that the adsorption and dissociation of H_2_O on the (110) surface can be completed in one microsecond (ms) at 298 K and in wet conditions; (3) the adsorption and dissociation of H_2_O on molybdenite are not the rate-limiting steps in the wet oxidation/weathering of molybdenite; and (4) kinetic Monte Carlo modeling explains the experimental SIMS observation that H_2_O and OH (rather than H^+^/H^−^ or H_2_O) occupy the surface of MoS_2_ in a short time. This study provides new molecular-scale insights to aid in our understanding of the oxidation/weathering mechanism of molybdenite as the predominant mineral containing molybdenum in the Earth’s crust.

## 1. Introduction

Molybdenum (4d^5^5s^1^ at the outermost shell) is a typical transition metal and exists mainly as Mo(VI) and Mo(IV) in nature. Therefore, molybdenum acts as an important tracer to construct the paleo-ocean condition due to its redox sensitivity to the aqueous environment [1,2]. Molybdenum is an essential trace element for organisms because it can form a series of important enzymes [3,4]. Its abundance in the Earth’s crust ranges from 0.6 to 1.4 mg/kg [5]. Molybdenite (MoS_2_) has been considered as the only mineral category with commercial significance, and it usually coexists with other sulfide minerals (e.g., pyrite, chalcopyrite) in the crust [6,7]. Most of the molybdenum minerals, such as molybdenite (MoS_2_), ferrimolybdite (Fe_2_(MoO_4_)_3_·nH2O), molybdite (CaMoO_4_), kamilokite (Fe_2_Mo_3_O_8_), wulfenite (PbMoO_4_) and drysdallite (Mo(S,Se)_2_), are nearly insoluble. Therefore, the weathering/oxidation of molybdenite is the fundamental geochemical process controlling the release of molybdenum from the lithosphere to the hydrosphere.

The oxidation of molybdenite (MoS_2_) cannot be monitored by Raman spectroscopy, even when exposed to pure oxygen and temperatures up to 340 °C [8]. However, the wet oxidation/weathering of molybdenite’s surface can be monitored by X-ray photoelectron spectroscopy (XPS) and atomic force microscopy (AFM) after it has been immersed in water for one hour [9] and exposed to humid air [10,11]. All these factors suggest that the adsorption of water molecules on the surface of molybdenite is a precondition for the weathering/oxidation of molybdenite [9]. However, the adsorption isotherm, site preferences and kinetics of water molecules on the different surfaces of molybdenite at the size and time scale of geochemical interest are still poorly understood.

A series of studies applying density functional theory (DFT) were performed to determine the adsorption site stability for H_2_O adsorption on the surface of the monolayer of molybdenite [12,13,14], because the adsorption and dissociation of H_2_O on the molybdenite surface is difficult to control and discriminate in experiments. According to the adsorption energy of water molecules at different sites, the adsorption site preferences can be compared [12,13,14]. The adsorption of H_2_O on the basal plane of the (001) monolayer of MoS_2_ shows a positive adsorption enthalpy change, indicating a repulsive interaction between the free H_2_O molecule and the perfect MoS_2_ (001) surface [13,14]. By comparing the adsorption energy of H_2_O above two types of S atoms (S atom with a single S-Mo bond and S atom with double S-Mo bonds) and a Mo atom, the adsorption of H_2_O above the S atom was considered unstable due to the repulsive interaction between the S and O atoms, while H_2_O adsorption above Mo was considered stable due to the attractive interaction between the O and Mo atoms [14]. Moreover, there are also other possible adsorption geometries to decrease the repulsive interaction between O and S atoms, e.g., the bond between H and S atoms instead of O and S atoms. Therefore, the adsorption site preference of H_2_O on the surface of MoS_2_ needs more detailed research. Theoretical modeling through first-principle calculation is a useful tool to explore the adsorption mechanism, but present studies are mainly focused on the mono-layer of MoS_2_ (001 surface). The surface properties largely depend on the size scale of the mineral surface. In a microscopic view, the interaction among the atoms in the long range can also affect the adsorption site. Therefore, there is always a difference in interest and scale when directly applying physical calculation to solve problems with more geochemical interest. For example, the natural mineral MoS_2_(2H) is a three-dimensional object and has three surfaces, i.e., (001), (110), (010). The lack of studies of the water adsorption mechanism (e.g., sorption isotherm, adsorption energy and kinetics) on these surfaces inhibits our fundamental understanding of the oxidation/weathering of molybdenite. To clarify these basic adsorption reactions regarding the interaction at the water–molybdenite surface, we employ annealing dynamics and kinetic Monte Carlo modeling to investigate these fundamental problems.

## 2. Method

### 2.1. Thermodynamic Calculation of Eh-pH Diagrams and Reaction Forward Modeling

To fully understand the effect of oxidation on the surface of molybdenite, an Eh-pH diagram is drawn based on the PHREEPLOT [15]. Moreover, forward reaction modeling is performed using the code PHREEQC [16].

### 2.2. Molybdenite Crystal and Geometry Optimization

Molybdenite mainly exists as 2H-MoS_2_ (space group: P63/mmc) in nature. To date, the reported molybdenite crystal categories include (1) the octahedral coordination of Mo (1T) and (2) the trigonal prismatic coordination of Mo (2H, 3R and 4H), which depends on the stacking sequences in S-Mo-S at the z-axis [17]. However, molybdenite in nature exists only as 2H-MoS_2_ (space group: P63/mmc) and 3R-MoS_2_ (space group: C5-R3m) [18,19,20]. Moreover, 2H-MoS_2_ is the predominant crystal category of molybdenite in the deposit [18,19,20]. Therefore, 2H-MoS_2_ (Figure 1) was selected for this study. The calculations in this study were performed using the code of Material Studio (Accelrys). To obtain the primary cell of 2H-MoS_2_, geometry optimization was performed using the Gaussian-Plane Wave (GPW) method. Different sets of functional and cutoff energy were comparatively used to calculate the lattice constant and band gap. The lattice constant, calculated using the functional PBE, successfully converged, and its result was closer to the experimental value than LDA and PW91 (Table 1). The functional PBE and energy cutoff of 398 eV were selected. The K-point was set as 5 × 5 × 1 in the calculations at the reciprocal space.

### 2.3. Sorption Isotherm

The dissociation of H_2_O into OH and H has been considered to mainly occur at the adsorption site above the Mo atom [13,14,24]. If the adsorption of H_2_O above Mo is the most stable adsorption site (with the lowest energy after adsorption), it is very difficult to explain the factors that drive the H_2_O above Mo to break the strong O-H bond and dissociate into OH and H. The adsorption of water molecules on the mineral surface is affected by the surface layers in the supercell, while the initial multi-atom system can result in exponential growth in the calculation cost and time. The universal force field (UFF) was used in the molecular dynamics modeling of H_2_O adsorbed on the (001) and (110) surfaces of molybdenite. UFF is constructed using the general rules only based on the element, its hybridization and its connectivity [25], and it has been widely used in calculating the interactions among atoms in molecular dynamics simulation. The 5 × 5 × 2 supercell of each surface, (001) and (110), was constructed in the calculation in order to more closely reflect the real size of molybdenite and consider the long-range interactions among atoms. At a constant temperature, average numbers of H_2_O adsorbed for each cell of the (001) and (110) surfaces of molybdenite were sampled and calculated from the atmospheric pressure from 1 bar to 10 bar. The temperature was increased from 273 to 398 K (0~120 °C) with a step of 25 K. The summation method was based on the Ewald Group and atom-based van der Waals. At each simulation step, the metropolis method was used to sample the ensembles. The probability density of the canonical ensemble is shown in Equation (1).
(1)$$\rho \left(\Gamma \right)=\frac{\exp\left(-\beta E\left(\Gamma \right)\right)}{\int d\Gamma \prime \exp\left(-\beta E\right))}$$
where *E(Γ)* is the potential energy (eV) of the system in state *Γ* and *β* = 1/(K_b_T), in which K_b_ is the Boltzmann constant and T is the thermodynamic temperature (K).

### 2.4. Kinetic Monte Carlo and Rate Coefficient Calculation

The most likely site and geometry for H_2_O adsorbed on the (001) and (110) surfaces of MoS_2_ were explored through the annealing process, which was realized by increasing the temperature of the system in steps to 100,000 K, and then automatically decreasing it to 100 K. In the molecular dynamics process, we also used the UFF, but with the Monte Carlo method to sample the energy of each state. The process was repeated in 3 cycles with 15,000 steps in each cycle to explore the state with lower energy.

Based on the Monte Carlo kinetics, rate activation energy and rate coefficient calculated, we investigated the dissociative adsorption rate of water at the mineral surface. Kinetic Monte Carlo (KMC) modeling is based on the rate coefficient or activation energy of each step reaction to simulate the concentration and rate change as a function of time, being comparable to laboratory settings. Its visualization result can help us to understand the reaction process at the atomic scale. According to the experimental observation of OH and H_2_O on the mineral surface [24] and the theoretical prediction of H desorption as H_2_ from the surface, in this study, KMC is employed to combine the three-stage reactions, which requires the rate coefficient or activation energy of H_2_O adsorption, H_2_O dissociation and H_2_ desorption. We did not consider the diffusion of these species at the surface, because the effect of diffusion on the overall reaction rate is not significant if the time scale concerned is in seconds or less. The adsorption rate coefficient of H_2_O adsorption on the (001) surface was calculated based on Equation (2) [26]:(2)$$K_{H_{2}O-\mathrm{ads}}=\frac{P_{H_{2}O\;}A_{\mathrm{site}}б}{\sqrt{2{\pi m}_{H_{2}O}K_{b}T}}$$
where K_H2O-ads_ is the rate coefficient for H_2_O adsorption at the surface of the mineral (s^−1^). A_site_ is the area of a single adsorption site (m^2^), which can be estimated based on the geometry of the cleaved surface in the calculation. PH_2_O is the pressure of water (atm). *б* is the sticking coefficient (s^−1^). mH_2_O is the molar mass of water molecules (g/mol). K_b_ is the Boltzmann constant (J/K) and T is the thermodynamic temperature (K).

The water vapor saturation pressure as a function of temperature is described with the Arden–Buck equation; see Equation (3) [27]. Its error compared to the experimental value [28] is less than 0.04% at the temperature range of 273.15 to 373.15 Kelvin.
(3)$$P=0.61121\;\exp\left[\left(18.678-\frac{\left(T-273.15\right)}{234.5}\right)*\left(\frac{\left(T-273.15\right)}{257.14+T}\right)\right]$$

P is the water vapor saturation pressure (KPa) and T is the thermodynamic temperature (K). Therefore, we can obtain the water adsorption rate coefficient reaching water vapor saturation at a certain temperature as Equation (4):(4)$$K_{H_{2}O-\mathrm{ads}}=\frac{0.00603\;\exp\left[\left(18.678-\frac{\left(T-273.15\right)}{234.5}\right)*\left(\frac{\left(T-273.15\right)}{257.14+T}\right)\right]A_{\mathrm{site}}б}{\sqrt{2{\pi m}_{H_{2}O}K_{b}T)}}$$
where K_b_ is the Boltzmann constant (J/K) and T is the thermodynamic temperature (K).

The rate coefficient of H_2_O dissociation at the (001) MoS_2_ surface has been reported previously [14]. The rate coefficient of H_2_ desorption from the (001) MoS_2_ surface has not been reported and no direct equation is available. The H_2_ desorption from the (001) MoS_2_ surface can be expressed as in Reaction 2. When obtaining the equilibrium between H_2_ adsorption and desorption on the (001) surface, the product of the H_2_ adsorption rate coefficient and H_2_ desorption rate coefficient will be equal to its equilibrium constant. Therefore, Equation (5) can be used to calculate the rate coefficient of H_2_ desorption from the (001) MoS_2_ surface (the detailed process is summarized in Appendix A as an elementary reaction.
2H-MoS_2_ = MoS_2_ + H_2_ (Reaction 2)
(5)$$K_{\mathrm{eq}}=\frac{K_{\mathrm{forward}}}{K_{\mathrm{reverse}}}=\frac{K_{H_{2}-\mathrm{desorption}}}{K_{H_{2}-\mathrm{adsorption}}}$$
where K_eq_ is the equilibrium constant of H_2_ adsorption and desorption at the (001) surface; K_forward_ and K_reverse_ are the rate coefficients (s^−1^) of the forward and reverse reaction, respectively; K_H2-desorption_ and K_H2-adsorption_ are the rate coefficients of H_2_ desorption and adsorption, respectively. The detailed calculation is presented in the Appendix A. Appendix A. The rate coefficient of H_2_ adsorption was also calculated using the same equation, Equation (2). Based on these parameters, the overall and step reaction rate in H_2_O’s dissociative adsorption as a function of time was predicted (Figure 2).

## 3. Results

### 3.1. Oxidation of Molybdenite Based on Thermodynamic Calculation

Based on the Eh-pH diagram of the Mo-S-H_2_O system at 25 °C and reference atmospheric pressure, we can find that MoS_2_ is stable under the acid and reductive conditions (Figure 2) and can be oxidized and transformed into Mo (VI) species in the surface environment. The oxidation energy difference varies from −0.9 to −2.4 eV based on the DFT calculation [29]. In acidic water (pH < 4.3) in the relatively oxidative condition, MoS_2_ can also remain stable (Figure 3). The forward path modeling of MoS_2_ oxidized by O_2_ indicates that oxidation will improve its solubility unlimitedly if there is sufficient O_2_/air and will decrease the pH (Figure 3). The increase in Mo (VI) and decrease in pH will form a series of aqueous species and complexes (Figure 3), such as MoO_4_^2−^, HMoO_4_^−^, H_2_MoO_4_, Mo_7_O_24_^6−^. All of these suggest that the oxidation of molybdenite is very important in understanding the Mo aqueous species during oxidation.
2MoS_2_ + 6H_2_O + 9O_2_ = 2MoO_4_^2−^ + 12H^+^ + 4SO_4_^2−^ (Reaction 1)

### 3.2. Sorption Isotherm of H_2_O on the (001) and (110) Surfaces

All of the movement of H_2_O at the surface of MoS_2_ is random in the simulation. The sorption result of H_2_O on the (001) and (110) surfaces of MoS_2_ from 1 bar to 10 bar at 298 K is presented (Figure 4). The sorption capacity comparison and the isotherms at different temperatures and pressures are presented (Figure 5 and Figure 6). According to the number of water molecules adsorbed on the supercells of the (001) and (110) surfaces from 1 to 10 bar (Figure 4), we can calculate the average H_2_O molecules adsorbed for each cell (Figure 5). H_2_O molecules adsorbed on the (001) surface are clearly distributed in different layers in the space, while those water molecules adsorbed on the (110) surface are relatively evenly distributed in the space (Figure 4). The average loadings of H_2_O molecules for the cells of the surfaces (001) and (110) at 298 K both increased with the H_2_O fugacity in the range of 1 to 10 bar (Figure 5). Moreover, the H_2_O molecule sorption capacity of the (110) surface at 298 K was much larger than that of the d(001) surface, and the capacity difference between the two surfaces also increased significantly with the H_2_O fugacity (Figure 5). Based on the average loading of H_2_O molecules sorbed on the cell and the stoichiometry mass of H_2_O and the cell, we can obtain the sorption isotherm from the (001) and (110) surfaces as a function of H_2_O fugacity and temperature (Figure 6). There were significant and highly positive linear relationships between the H_2_O capacity on the two surfaces and the H_2_O fugacity (Figure 6). When the temperature increased from 273 to 398 K, the H_2_O sorption capacity on both the (001) and (110) surfaces decreased significantly, as shown by the decrease in slope for each isotherm line at the two surfaces (Figure 6). The sorption capacity of H_2_O on the (110) surface was stronger than that of the (001) surface at all temperatures and H_2_O fugacity values investigated (Figure 6).

### 3.3. Adsorption Energy Derived from the Annealing Dynamics of H_2_O on the (001) and (110) Surfaces

The annealing dynamics from 100,000 K to 100 K indicated that there were two possible adsorption sites on the (001) surface, 001-G1 and 001-G2, whose adsorption energy was −2.51 and −2.23 eV, respectively (Figure 7), and there were four possible adsorption sites on the (110) surface—110-G1, 110-G2, 110-G3 and 110-G4—whose adsorption energy was −2.56, −2.36, −2.15, −1.93 and −1.48 eV, respectively (Figure 7). The stability of different adsorption sites on the surfaces can be reflected by the potential energy. Correspondingly, the stability of the adsorption sites on molybdenite follows the order 110-G1 > 110-G2 > 001-G1 > 001-G2 > 110-G3 > 110-G4 > 110-G5, as shown by the increase in negative adsorption energy for H_2_O in the order (Figure 7). Moreover, the four most stable geometries of H_2_O adsorption sites are presented (Figure 8 and Figure 9). To clarify the geometry on the sites, the most stable adsorption sites on the (001) surface were (1) 001-G1, H_2_O above the S atom at the edge; (2) 001-G2, H_2_O above the S atom closing the edge. Both 001-G1 and 001-G2 were above the S atoms, with two H atoms oriented towards the S atom. The two most stable adsorption sites on the (110) surface were (1) 110-G1, H_2_O above the S-Mo bond at the edge, with two H atoms oriented towards the S atom and an O atom oriented towards the Mo atom (Figure 8); (2) 110-G2, H_2_O above the S-Mo bond close to the edge, with two H atoms oriented towards the S atom and an O atom oriented towards the Mo atom (Figure 9). The distance between two H atoms and an S atom for 001-G1, 001-G2, 110-G1 and 110-G2 was 3.311–3.313 3.128–3.198, 3.382–3.392, 3.326–3.355 Å, respectively, and only 001-G1 had two equal S-H distances (Figure 8 and Figure 9). The specific distances between O and H atoms from H_2_O and Mo and H atoms from MoS_2_ for the four most stable sites are presented (Figure 8 and Figure 9). The distance between the O atom and Mo atom of 001-G1 and 001-G2 could not be determined due to the uncertain direction and overly long distance (Figure 8).

### 3.4. Adsorption Rate Coefficient of H_2_O on the (001) and (110) Surfaces as a Function of the Temperature and Humidity

Based on Equation (2), we can calculate the adsorption rate coefficient of H_2_O on the mineral surface at certain partial pressure and temperature values; the A_site_ is the area of a single sorption site, which equals the reciprocal of the sorption site density. According to the area of the basal plane and the H_2_O adsorption number per cell, we can calculate the area of a single sorption site at the (001) and (110) surfaces, respectively (Table 2). Combined with the bulk equation, we can calculate the saturation vapor pressure of H_2_O at certain temperature ranges from 0 to 100 °C (Figure 10a) and calculate the corresponding adsorption rate coefficient at certain conditions. The adsorption rate coefficient of H_2_O on the (110) surface is close to that on the (001) surface at a temperature from 0 to 25 °C, but it increases more significantly than the (001) surface with the temperature. When reaching 100 °C, the adsorption rate coefficient of H_2_O on the (110) surface is approximately 16 times higher than that on the (001) surface. To explore the impact of humidity on the adsorption of H_2_O on MoS_2_, the adsorption rate coefficient of H_2_O on the (001) and (110) surfaces of MoS_2_ at 25 °C was presented as a function of relative humidity (Figure 10b). The adsorption rate coefficients of H_2_O on the (110) and (001) surfaces both increased with the humidity.

### 3.5. Kinetics of H_2_O Dissociative Adsorption on the (110) Surface

The adsorption mechanism and rate coefficient both indicate that H_2_O adsorption on the (110) surface is always favored at all of the temperature and humidity levels investigated. Therefore, the surface reactions of H_2_O on the (110) surface are selected in the KMC modeling, which includes the kinetics of (1) the adsorption of H_2_O on the surface; (2) the dissociation of H_2_O into OH and H on the surface; (3) the desorption of H as H_2_ molecules from the surface. According to Equations (2) and (3), we can obtain the rate coefficient of the three-stage reaction. When reaching the condition with 20% relative humidity of H_2_O in air at 298 K, the adsorption rate coefficient is 4.84 × 10^7^ S^−1^. The rate coefficient of H_2_O dissociation on the (110) surface of MoS_2_ at 300 K is 2.8 × 10^10^ S^−1^. The rate coefficient of H_2_ desorption on MoS_2_ is 2.38 × 10^8^ S^−1^ (detailed calculation is presented in Appendix A.). To more closely approach the real situation, the rate coefficient of H_2_O desorption from MoS_2_ was also calculated (detailed calculation is presented in Appendix A). The metadynamic calculation provides the rate coefficient of H_2_O dissociation on the MoS_2_ surface [14]. Based on the reaction free energy of the dissociation, we can calculate the reverse reaction (the formation of H_2_O from dissociated OH and H) rate coefficient as 2.9 × 10^11^ S^−1^ (detailed calculation is presented in Appendix A). The average H_2_O number adsorbed on the (110) surface is approximately 0.85 per cell at 298 K at the water vapor saturation pressure. Therefore, we constructed a surface composed of 32 × 32 adsorption sites, which represents a (110) surface composed of 37 × 37 cells, to predict the overall surface reactions as a function of time in seconds.

Reaction scheme in kinetic Monte Carlo and rate coefficient are presented in Table 3. Based on the reaction rate coefficient of these reactions, the kinetic model of the surface reaction on the (110) surface shows that H_2_O is firstly adsorbed on the surface and 20% of the adsorption sites are occupied by H_2_O, and only a few OH and H atoms appear at the surface at 10–20 ns (Figure 11 and Figure 12). The concentration of H_2_O adsorbed on the (110) surface increases at a reaction time of less than 40 ns, while it gradually decreases afterwards. The concentration of OH on the surface increases rapidly with time and reaches approximately 90% at a reaction time of 1 ms (Figure 11 and Figure 12). The concentration of H adsorbed on the surface is very low and decreases to nearly zero in 1 ms (Figure 11 and Figure 12).

## 4. Discussion

### 4.1. Adsorption Capacity and Rate Comparison of H_2_O on the Different Surfaces of Molybdenite and Its Implications for the Mineral Reactivity

The distribution of H_2_O molecules in the space above the surface is different between surfaces (001) and (110), indicating that the sorption capacity and mechanism of H_2_O on the two surfaces are significantly different (Figure 4). The H_2_O adsorption on the (001) surface is multi-layer adsorption, while the H_2_O adsorption on the (110) surface is single-layer adsorption (Figure 4). The periodic adsorption of H_2_O above the (001) surface (Figure 4) suggests that it is probably the result of the periodic repulsive force between the O atom of H_2_O and the S atom of MoS_2_. The sorption capacity of H_2_O on the two surfaces increases with the H_2_O fugacity but decreases with the temperature (0–120 °C), indicating that the relatively wet and cold environment is beneficial for the H_2_O molecules’ adsorption capacity on the mineral surface (Figure 7), while a very low temperature can decrease the adsorption rate of H_2_O on the surface (Figure 10). The sorption isotherm of H_2_O shows that the sorption capacity of the (110) surface is much stronger than that of the (001) surface (Figure 5 and Figure 6), indicating that the (110) surface can obtain more water adsorbed on its surface in the weathering of MoS_2_ and its interaction with water.

Although a temperature decrease can improve the sorption capacity of H_2_O on the MoS_2_ surface, it can significantly decrease the adsorption rate of H_2_O on the MoS_2_ surface, suggesting that the effect of temperature on water sorption on MoS_2_ is complicated. The adsorption rate coefficient of H_2_O on the (110) surface was always larger than that on the (001) surface at all the temperatures (0~100 °C) and relative humidity (0~100%) investigated (Figure 10). Therefore, both the adsorption capacity and rate coefficient of H_2_O on the (110) surface are stronger than those on the (001) surface, suggesting that the (110) surface is the predominant surface that controls the solubility and reactivity of MoS_2_ in its interaction with water, because the (110) surface can attract many more water molecules, and the atoms of Mo and S and their bonds are also exposed to the H_2_O molecules.

### 4.2. Adsorption Mechanism of H_2_O on the Surfaces of Molybdenite

Compared to the possible adsorption geometry optimization using DFT techniques at 0 K, the annealing dynamics method explores many more possible adsorption geometries at a larger temperature range from 10 to 100,000 K. The most stable adsorption sites of H_2_O on the (001) and (110) surfaces derived from annealing dynamics are much more stable than the sites derived from optimization with DFT. For example, the most stable adsorption site on the Mo and S atoms at the edge of the (001) surface (equivalent to the (110) surface) has the lowest adsorption energy of −0.55 eV [14], which is much higher than that of the adsorption sites with adsorption energy lower than −2.5 eV, e.g., 001-G1 and 110-G1 (Figure 7). The adsorption sites of H_2_O on the (001) surface have been considered unstable (positive adsorption energy) due to the repulsive interaction between S and O atoms [14], but the annealing dynamics indicated that H2O molecules adsorbed on the (001) surface can also remain stable (e.g., 001-G1 and 001-G2 with negative adsorption energy lower than −2.2 eV) by means of changing the direction of H_2_O molecules to direct the two H atoms towards the S atom at the surface in order to lower the total energy and repulsive interaction (Figure 8). For the (110) surface, the adsorption site 001-G1 has larger distances between Mo and O atoms and between S and H atoms than 001-G2, which causes the O atom (of H_2_O) and S atom (of MoS_2_) to receive more electrons from the Mo atom (of MoS_2_) and H atom (of H_2_O). However, the 001-G1 adsorption site is more stable than 001-G2, as shown by the lower adsorption energy (Figure 7), because 001-G1 has larger distances between O and S and between H and Mo atoms (Figure 9b,d), suggesting that the repulsive interactions between O and S and between H and Mo atoms are the major factors that control the energy level of the adsorption geometry. These results suggested that the annealing dynamics and larger supercells with more periodic clusters can provide more possible and stable sites than the common cluster optimization, which provides new, complete insights in understanding the adsorption mechanism at the level of geochemical interest.

### 4.3. Kinetic Modeling of Dissociative Adsorption of H_2_O on the (110) Surface and Its Implication for the Weathering of Molybdenite in the Surface Environment

We selected the (110) surface to predict the progress of H_2_O adsorption and dissociation as a function of time due to the great reactivity of the (110) surface of molybdenite. Although a previous study showed that the H_2_O adsorbed on molybdenite can dissociate into OH and H at room temperature [15], the effects of the adsorption/desorption rate of water, the formation rate of OH and H into H_2_O and the desorption and sorption rate of H on the overall reaction rate have not been considered. To more closely reflect the real situation, we calculated and considered the reaction rates of all of these step reactions in the KMC modeling, which was firstly applied to predict the dissociation adsorption of the H_2_O reaction on the (110) surface as a function of time (Figure 11 and Figure 12). The modeling result indicated that the adsorption and dissociation of H_2_O into OH and H can occur in the time scale of ns (10^−9^ s) at 25 °C and in a humid environment (Figure 11 and Figure 12). Moreover, the OH can occupy most adsorption sites of the (110) surface in one microsecond (Figure 11 and Figure 12), suggesting that OH covers the surface of MoS_2_ in a very short time, and the dissociative adsorption of H_2_O is not the rate-limiting step in the oxidation of MoS_2_ at a normal temperature. This explains the experimental SIM observation that H_2_O and OH (rather than H^+^/H^−^ or H_2_) occupied the surface of MoS_2_ in a short time [24]. In the overall reaction of MoS_2_ oxidation (Reaction 1), the adsorption/capture of molecular oxygen on the OH shell around molybdenite may be an important reaction controlling the rate of MoS_2_ oxidation/weathering, which needs further detailed investigation.

## 5. Conclusions

By employing molecular dynamics and kinetic Monte Carlo modeling, we investigated the adsorption isotherm of H_2_O on the (001) and (110) surfaces of molybdenite at temperatures ranging from 0 to 120 °C and H_2_O fugacity ranging from 1 to 10 bar, which indicated that an increase in fugacity and decrease in temperature can promote adsorption, but a very low temperature can lower the adsorption rate coefficient. The adsorption geometry of H_2_O on the (001) and (110) surfaces at 298 K was explored using annealing dynamics, which provides a more stable geometry with lower adsorption energy, relative to a previous report using DFT geometry optimization. The adsorption rate coefficient and capacity of H_2_O on the two surfaces indicated that the (110) and (010) surfaces are the predominant surfaces that control the solubility and reactivity of MoS_2_ in its interaction with water. The reaction time of 1 ms (10^−3^ s) is sufficient for H_2_O to adsorb and dissociate into OH at the (001) surface, suggesting that the adsorption and dissociation of H_2_O on the MoS_2_ surface is not the rate-limiting step in MoS_2_ oxidation/weathering.

## Figures and Tables

**Figure 1 molecules-27-08710-f001:**
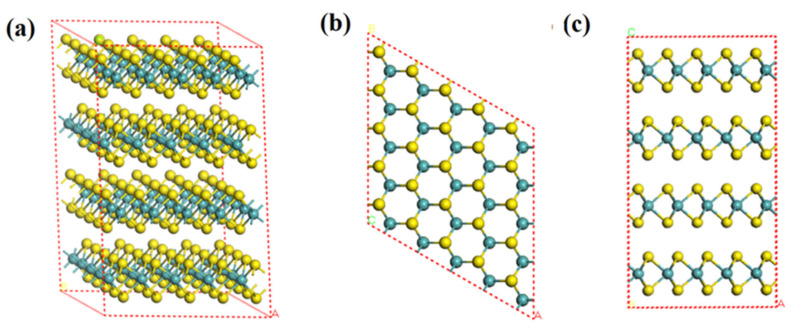
Supercell of MoS_2_(2H) crystal (5 × 5 × 2) (**a**): three-dimensional, (**b**): top view (001), (**c**): side view (110) (yellow ball is sulfur atom and blue ball is molybdenum atom).

**Figure 2 molecules-27-08710-f002:**
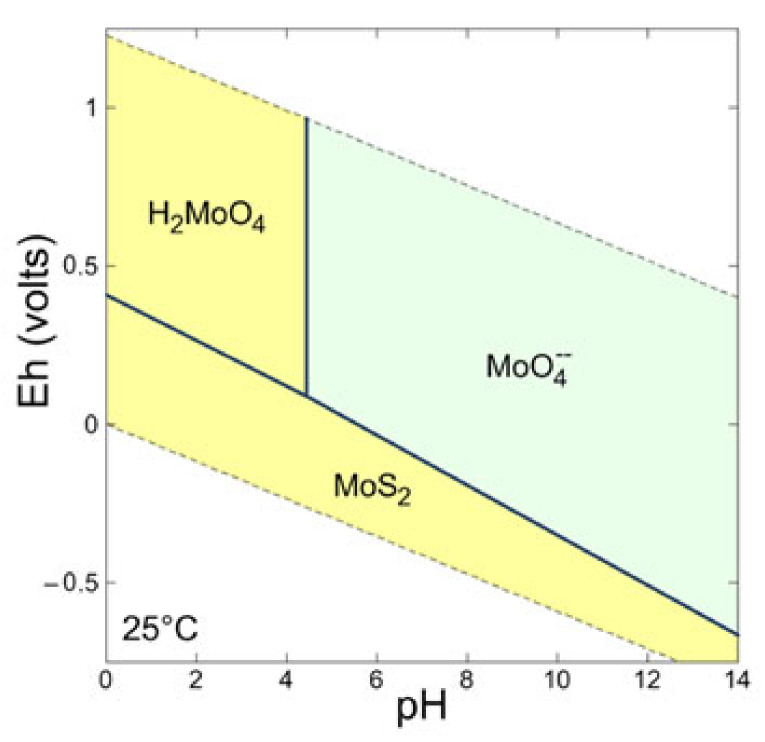
Eh-pH diagram of Mo-S-H_2_O system at 25 °C and 1 bar.

**Figure 3 molecules-27-08710-f003:**
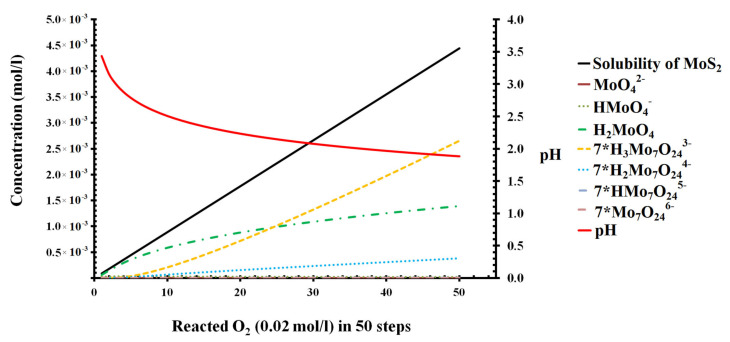
Oxidation process of molybdenite and speciation change as a function of the number of reaction steps.

**Figure 4 molecules-27-08710-f004:**
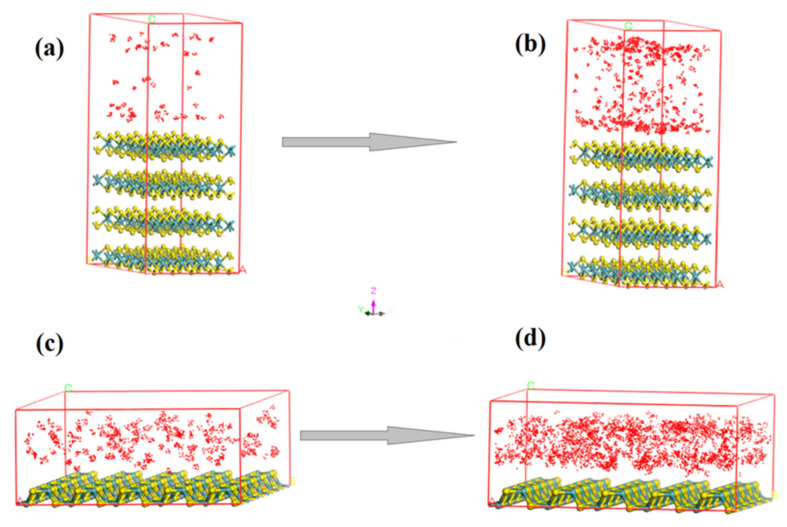
Sorption saturation status of H_2_O molecule on the supercell (5 × 5 × 2) of (001) surface at (**a**) 1 bar and (**b**) 10 bar, and (110) surface at (**c**) 1 bar and (**d**) 10 bar (one red point represents one water molecule; yellow ball represents sulfur atom; blue ball represents molybdenum atom).

**Figure 5 molecules-27-08710-f005:**
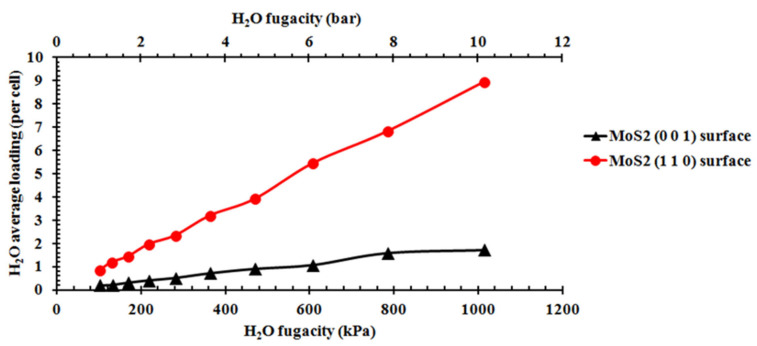
Sorption capacity change of H_2_O on the (001) and (110) surfaces of MoS2 at 1 and 10 bar.

**Figure 6 molecules-27-08710-f006:**
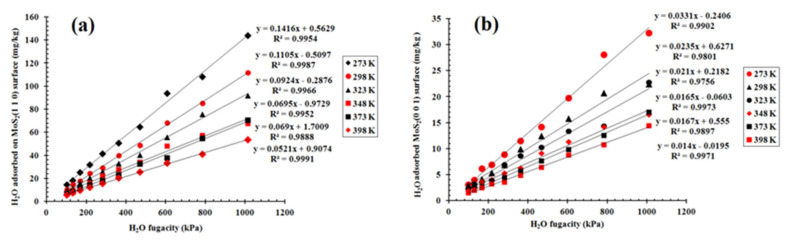
Sorption isotherm of H_2_O molecules on the (**a**): (001) and (**b**): (110) surfaces as a function of temperature.

**Figure 7 molecules-27-08710-f007:**
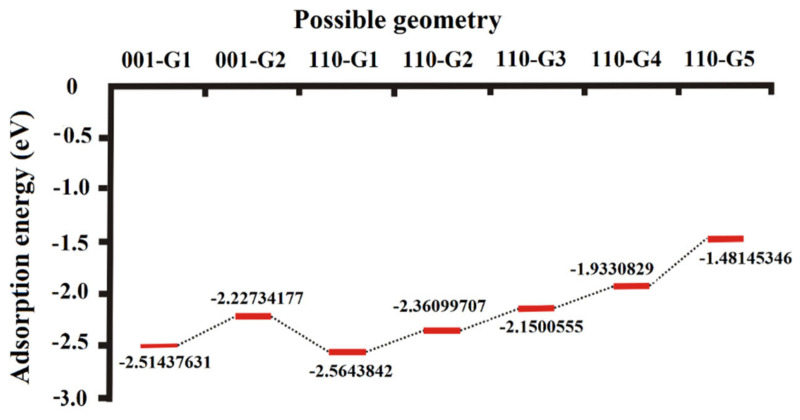
Adsorption energy of H_2_O at different adsorption sites on the (001) and (110) surfaces.

**Figure 8 molecules-27-08710-f008:**
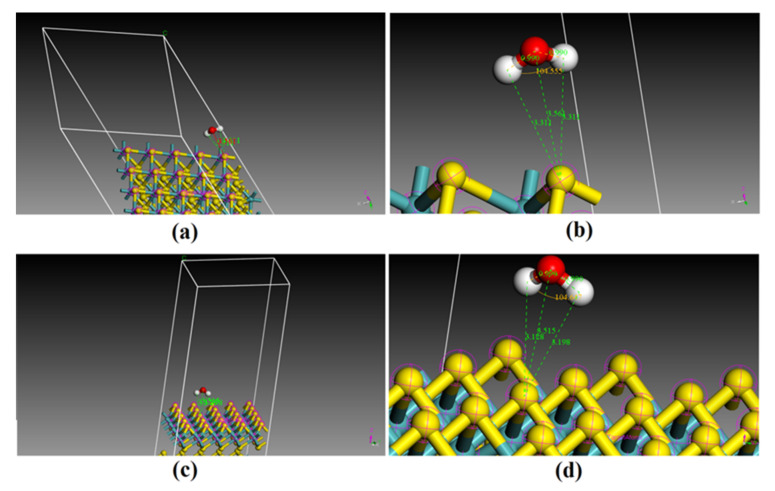
Geometry of the two most stable adsorption sites for H_2_O on the (001) surface: (**a**,**b**): 001-G1; (**c**,**d**): 001-G2 (yellow ball represents sulfur atom; blue ball represents molybdenum atom).

**Figure 9 molecules-27-08710-f009:**
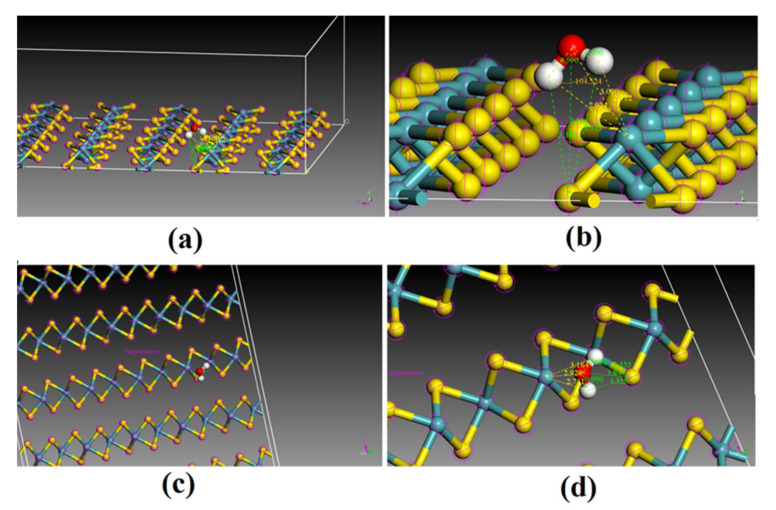
Geometry of the two most stable adsorption sites for H_2_O at the (110) surface: (**a**,**b**): 110-G1; (**c**,**d**): 110-G2 (yellow ball represents sulfur atom; blue ball represents molybdenum atom).

**Figure 10 molecules-27-08710-f010:**
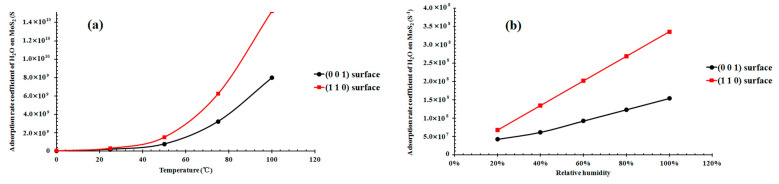
Adsorption rate coefficient of water molecules on molybdenite as a function of temperature (**a**) and relative humidity (**b**).

**Figure 11 molecules-27-08710-f011:**
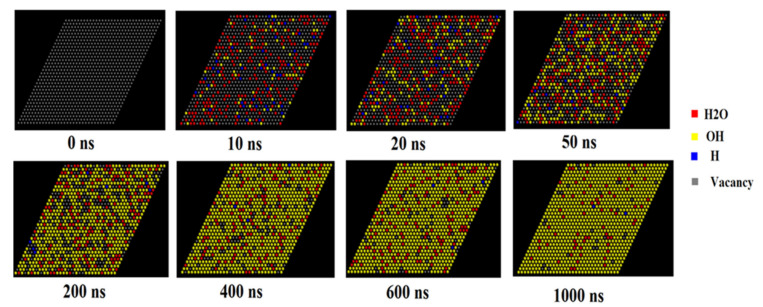
KMC modeling of the dissociative adsorption of H_2_O on the molybdenite (001) surface in one microsecond at 298 K.

**Figure 12 molecules-27-08710-f012:**
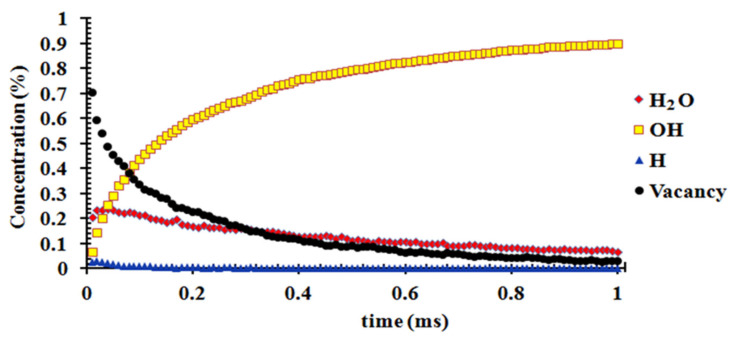
Relative concentration variance of H_2_O, OH, H on the molybdenite (110) surface in one microsecond at 298 K.

**Table 1 molecules-27-08710-t001:** Comparison of DFT calculation using different functional and experimental parameters in MoS2 cell geometry optimization.

Source	Lattice Constant—a (Å)	Lattice Constant—c (Å)	Mo-S Bond Length (Å)	S-S Bond Length (Å)	Band Gap (eV)
LDA	3.136	12.052	2.381	3.110	0.74
PBE	3.168	12.616	2.407	3.128	1.102
PW91	3.180	12.690	2.411	3.126	1.023
Dickinson and Pauling, 1923 [21]	3.15	12.3			1.29
Swanson et al., 1955	3.16	12.295			1.29
Bronsema et al., 1986 [22]	3.16	12.296			
Zubavichus et al., 1998 [23]			2.39	3.16	

**Table 2 molecules-27-08710-t002:** Adsorption rate coefficient of H_2_O (K_H2O-ads_) on the (001) and (110) surfaces of molybdenite at temperature (t) from 0 to 100 °C and corresponding water vapor saturation pressure (P_vp_).

	T (°C)	P_vp_ (atm)	A-Site (Å2)	K_H2O-ads_ (S^−1^)
(001)	0	0.006	3.23	2.97 × 10^7^
	25	0.0313	4.63	2.13 × 10^8^
	50	0.1218	4.57	7.84 × 10^8^
	75	0.3806	6.27	3.24 × 10^9^
	100	1	6.09	7.98 × 10^9^
(110)	0	0.006	5.68	5.22 × 10^7^
	25	0.0313	7.31	3.35 × 10^8^
	50	0.1218	8.92	1.53 × 10^9^
	75	0.3806	12.10	6.25 × 10^9^
	100	1	11.61	1.52 × 10^10^

**Table 3 molecules-27-08710-t003:** Reaction scheme in kinetic Monte Carlo dynamics and rate coefficient.

Reaction	Rate Coefficient (S^−1^)
MoS_2_ + H_2_O = MoS_2_-H_2_O	4.84 ×10^7^
MoS_2_-H_2_O = MoS_2_ + H_2_O	3.62 × 10^10^
MoS_2_-H_2_O = MoS_2_-OH + H	2.8 × 10^10^
MoS_2_-OH + H = MoS_2_-H_2_O	2.9 × 10^11^
2H-MoS_2_ = MoS_2_ + H_2_	2.38 × 10^8^
MoS_2_ + H_2_ = 2H-MoS_2_	1.20 × 10^8^

## Data Availability

All the data used is included in the article and Appendix A.

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
