# Peer review of "Adsorption Equilibrium and Mechanism and of Water Molecule on the Surfaces of Molybdenite (MoS2) Based on Kinetic Monte-Carlo Method"

_molecules, 2022, doi:10.3390/molecules27248710_

Round 1

Reviewer 1 Report (New Reviewer)

The manuscript “Adsorption equilibrium and mechanism and of water molecule on the surfaces of molybdenite (MoS2) based on kinetic Monte-Carlo method” by  Hang Zhang, Ruilin Wang, Xinyu Wang, Zhijun Zuo, Shijun Ni, Jie Dai and Zeming Shi theoretically investigates the weathering of molybdenite. Although the manuscript contains some interesting scientific findings I cannot recommend it for publication.

Below I listed several comments that can serve as a guidance for the authors to enhance the quality of the manuscript.

1.       The manuscript is badly written. Even the title is impossible to comprehend. The text contains endless typographic errors, badly formulated sentences, and unclear phases. All of that results in the inability of the reader to understand the scientific part of the manuscript and thus evaluate its quality. I will not list here all the examples of bad grammar, too many of them are present in the text. I suggest the authors rewrite the manuscript completely and then ask a native English language speaker to proofread the text. When the text is correct, I believe, the scientific part will become clear to the readers.

2.       The authors must clearly distinguish between methods and results. In the current version of the manuscript, Section 2 contains both, method description and calculation results as exemplified by Figure 4. Section “Methods” must contain the description of the methods only without any figures. The methods’ description must be full and unambiguous, if the authors employed theoretical calculations they should specify the version of the software, the type of the algorithm and the calculation parameters that they used. For the kinetic Monte Carlo, it is not the case. Did the authors use some commercial software?   

3.       When describing the kinetic Monte Carlo simulations authors first mention 3 reactions (line 305), but later they mention that the dissociation of water was reversible. Which reaction scheme did they use for the calculations? Which reactions were reversible? Why? Why didn’t they consider the sorption of the water molecule to be reversible?

4.       When calculating the kinetic rate coefficients in section 2.4 the authors use both “t” and “T” symbols for temperature. And sometimes both symbols are present in the equation. What do these symbols attribute to? I suggest that the authors clean up the equations and define each symbol there.

5.       The layout of the manuscript must be corrected as well. Sometimes authors write “Kinetic Monte-Carlo”, and sometimes “Kinetic Monte Carlo”; sometimes indexes in the chemical formulas are subscripts, sometimes not. All these inaccuracies hinder the scientific part of the paper even more.

After the authors solve the above-mentioned issues with this manuscript, I encourage them to resubmit this contribution.     

Author Response

Reviewer 1

The manuscript “Adsorption equilibrium and mechanism and of water molecule on the surfaces of molybdenite (MoS2) based on kinetic Monte-Carlo method” by  Hang Zhang, Ruilin Wang, Xinyu Wang, Zhijun Zuo, Shijun Ni, Jie Dai and Zeming Shi theoretically investigates the weathering of molybdenite. Although the manuscript contains some interesting scientific findings I cannot recommend it for publication.

Below I listed several comments that can serve as a guidance for the authors to enhance the quality of the manuscript.

  1. The manuscript is badly written. Even the title is impossible to comprehend. The text contains endless typographic errors, badly formulated sentences, and unclear phases. All of that results in the inability of the reader to understand the scientific part of the manuscript and thus evaluate its quality. I will not list here all the examples of bad grammar, too many of them are present in the text. I suggest the authors rewrite the manuscript completely and then ask a native English language speaker to proofread the text. When the text is correct, I believe, the scientific part will become clear to the readers.

Response: Thank you very much for your advices and the manuscript and title have been carefully rewritten in the revised manuscript. I am sorry for the grammar problems. Moreover, we also invited a native English speaker to improve the English of this manuscript.

  1. The authors must clearly distinguish between methods and results. In the current version of the manuscript, Section 2 contains both, method description and calculation results as exemplified by Figure 4. Section “Methods” must contain the description of the methods only without any figures. The methods’ description must be full and unambiguous, if the authors employed theoretical calculations they should specify the version of the software, the type of the algorithm and the calculation parameters that they used. For the kinetic Monte Carlo, it is not the case. Did the authors use some commercial software?   

Response: Thank you very much. The reason we put figure names in the section of method is wish to help readers to clearly reconstruct the calculation and figures. But the section relating to the results in the method has been moved to the result section in the revised manuscript (lines 191-194 in the revised manuscript). We employ the kinetic Monte Carlo in the commercial software “Material Studio (Accelrys)” which is also clearly presented in the manuscript. Our affiliation “ Laboratory of Coal Science and Technology of Ministry of Education and ShanxiProvince, Taiyuan University of Technology”has paid for this code. And the code name we used is also presented in the manuscript (lines 109 in the revised manuscript).

  1. When describing the kinetic Monte Carlo simulations authors first mention 3 reactions (line 305), but later they mention that the dissociation of water was reversible. Which reaction scheme did they use for the calculations? Which reactions were reversible? Why? Why didn’t they consider the sorption of the water molecule to be reversible?

Response: Thanks. The desorption rate and reversibility of water molecule is considered in KMC modeling in the manuscript. And the desorption rate calculation is also included in the supplementary materials along with the revised manuscript.

  1. When calculating the kinetic rate coefficients in section 2.4 the authors use both “t” and “T” symbols for temperature. And sometimes both symbols are present in the equation. What do these symbols attribute to? I suggest that the authors clean up the equations and define each symbol there.

Response: Thank you for your advices. The temperature is uniformly presented with thermodynamic temperature (T) in section 2.4 of the revised manuscript (line 169 in the revised manuscript).

  1. The layout of the manuscript must be corrected as well. Sometimes authors write “Kinetic Monte-Carlo”, and sometimes “Kinetic Monte Carlo”; sometimes indexes in the chemical formulas are subscripts, sometimes not. All these inaccuracies hinder the scientific part of the paper even more.

Response: Thanks for your careful review and we used the “Kinetic Monte-Carlo” in the revised manuscript to replace another usage (e.g., line 151 in the revised manuscript). And we also carefully checked the terms.

Reviewer 2 Report (New Reviewer)

Title: Adsorption equilibrium and mechanism and of water molecule on the
surfaces of molybdenite (MoS2) based on kinetic Monte-Carlo method

The authors have done nice work and in my opinion manuscript can be accepted after minor revision

My comments are following

·         Please improve abstract section as it is not well written.

·         Captions under the Figures are not listed properly, please check it carefully

·         It’s better to add a Table at the end of manuscript having all abbreviations used in manuscript

·         It is suggested to extend discussion about Adsorption site preference in section 2.3.

·         Please write chemical formulas properly according to standard format as it should be H2O not H2O as written in manuscript at line 225.

Author Response

Response to the comments of reviewers

Reviewer 2

Title: Adsorption equilibrium and mechanism and of water molecule on the
surfaces of molybdenite (MoS2) based on kinetic Monte-Carlo method

The authors have done nice work and in my opinion manuscript can be accepted after minor revision

My comments are following

  • Please improve abstract section as it is not well written.

Response: Thanks. The abstract has been rewritten. The order of the abstract and expression has been clearly presented (lines 15 to 29 in the revised manuscript).

  • Captions under the Figures are not listed properly, please check it carefully

Response: Thank you. All the figure captions are carefully checked.

  • It’s better to add a Table at the end of manuscript having all abbreviations used in manuscript

Response: Thanks. We added a table including all abbreviations (Table 3).

  • It is suggested to extend discussion about Adsorption site preference in section 2.3.

Response: Thanks. The section 2.3 has been added to section 2.4 in the revised manuscript.

  • Please write chemical formulas properly according to standard format as it should be H2O not H2O as written in manuscript at line 225.

Response: Thanks. We have revised the term (line 190 in the revised manuscript).

Reviewer 3 Report (New Reviewer)

The manuscript presents a study of water adsorption on molybdenite surfaces.
Sorption capacities, isotherms, and rates are calculated, and the adsorption energies
at different surface sites are determined. The study helps to understand the general
mechanisms of water adsorption on that mineral surface, with possible relevance to
geoscience studies. Thus, I consider that the scientific content of the paper is good.

However, the presentation of the paper is not good. Wording is poor.
There is a supplement of the manuscript, but this supplement was not made available.
The quality of some illustrations is the only positive aspect of the presentation.
Thus, the manuscript needs significant changes to be published. I would recommend the
authors to ask help of someone with practice in scientific writing in English language.

My main criticisms are:
1) The text in lines 45-55 has repetitions (about Mo in the crust). The comments on Fig. 1
are confusing in lines 56-66. Informal terms such as "parameters like x, y, z" should not
be used.
2) The references from which figs. 1 and 2 were obtained should be cited in the captions.
3) Page 4 contains many technical details on previous works on the subject. They should
be summarized because that is an Introduction. Possibly the details can be presented in
the Methods section as e.g. "relations with other works".
4) The first paragraph of sec. 2.1 has several repetitions and states that "more details
are included in section 2.1". This shows how the text is disorganized.
5) The paragraph in lines 134-143 is not clear. If the method requires an lenghty
explanation, then it should be presented in a supplement.
6) The first paragraph of Sec. 2.2 also has some confusing statements. Moreover, it mixes
results (Figs. 4, 5, and 6) with the Methods section, which is not appropriate. The results
should be presented only in sec. 3 to avoid confusion for the reader.
7) In figs. 5 and  10, I see no reason for connecting the data with lines.
8) Eqs. 3 and 4 should be written with square brackets instead of two levels of parenthesis.
9) Reference to R3 is not clear in line 210. Do you mean eq. 3?
10) There are repetitions in the lines around eq. 3 and they refer to a supplement that
was not attached to the review material.
11) In fig. 6, why is the adsorbed amount given in mg/kg? I would expect a mass per unit
area for such a surface process.
12) In sec. 3.2, to justify the results in Fig. 7, it is necessary to show the sorbed content
as a function of the temperature (I guess these are the results from which the data were
obtained).
13) Lines 287-289: maybe you can see differences between 0 and 25 oC using log scale.
14) In line 347, why a reference to fig. 7? In line 377, do you mean fig. 8 or fig. 9?

Author Response

Response to the comments of reviewers

Reviewer 3

The manuscript presents a study of water adsorption on molybdenite surfaces.
Sorption capacities, isotherms, and rates are calculated, and the adsorption energies
at different surface sites are determined. The study helps to understand the general
mechanisms of water adsorption on that mineral surface, with possible relevance to
geoscience studies. Thus, I consider that the scientific content of the paper is good.
However, the presentation of the paper is not good. Wording is poor.
There is a supplement of the manuscript, but this supplement was not made available.
The quality of some illustrations is the only positive aspect of the presentation.
Thus, the manuscript needs significant changes to be published. I would recommend the
authors to ask help of someone with practice in scientific writing in English language.
Response: Thank you for your review and we carefully revised the manuscript according to your advices.
My main criticisms are:
1) The text in lines 45-55 has repetitions (about Mo in the crust). The comments on Fig. 1
are confusing in lines 56-66. Informal terms such as "parameters like x, y, z" should not
be used.

Response: Thank you very much for your advice. The lines 45-55 in the original manuscript have been shorten as illustrated (lines 56 in the revised manuscript). Meantime, the parameters like x, y, z and the reaction are deleted to help readers to easy understand (line 56 in the revised manuscript). 

2) The references from which figs. 1 and 2 were obtained should be cited in the captions.

Response: The Fig. 1 and 2 are the original content of this manuscript. Therefore, we don’t have citation in the figure captions.
3) Page 4 contains many technical details on previous works on the subject. They should
be summarized because that is an Introduction. Possibly the details can be presented in
the Methods section as e.g. "relations with other works".

Response: Thank you. This section relating with technical details has been moved to Method section (lines 119 to 124 in the revised manuscript). 
4) The first paragraph of sec. 2.1 has several repetitions and states that "more details
are included in section 2.1". This shows how the text is disorganized.

Response: Thanks. The sentence has been deleted and the text of section 2.1 is reorganized in the revised manuscript.
5) The paragraph in lines 134-143 is not clear. If the method requires an lenghty
explanation, then it should be presented in a supplement.

Response: Thanks. We shorten this description and keeps it still in the method because the the basis sets and parameters are important. 
6) The first paragraph of Sec. 2.2 also has some confusing statements. Moreover, it mixes
results (Figs. 4, 5, and 6) with the Methods section, which is not appropriate. The results
should be presented only in sec. 3 to avoid confusion for the reader.

Response: Thank you for your advice. We moved the Figs. 4, 5 and 6 to the result section 3.
7) In figs. 5 and 10, I see no reason for connecting the data with lines.

Response: Thanks. The lines in Fig. 5 and 10 is used to help readers to clearly predict the values if needed between these points.
8) Eqs. 3 and 4 should be written with square brackets instead of two levels of parenthesis.

Response: Thanks. We changed as you suggested (lines).
9) Reference to R3 is not clear in line 210. Do you mean eq. 3?

Response: Sorry. R3 means the reaction 3 in the original manuscript. We changed the term to avoid misunderstanding. And the reaction 3 has changed to reaction 2 (line 181 in the revised manuscript).
10) There are repetitions in the lines around eq. 3 and they refer to a supplement that
was not attached to the review material.

Response: Thanks. The supplementary material is attached along with the revised manuscript.
11) In fig. 6, why is the adsorbed amount given in mg/kg? I would expect a mass per unit
area for such a surface process.

Response: Thanks. We use this unit to avoid more parameters, because the surface area is usually expressed in m2/kg and the adsorbed amount is used as g/m2. Therefore, the unit g/kg is more convenient to use for readers.
12) In sec. 3.2, to justify the results in Fig. 7, it is necessary to show the sorbed content
as a function of the temperature (I guess these are the results from which the data were
obtained).

Response: Thanks. Actually, the Fig. 6 provides the sorbed content as the function of the temperature. And they all used the same data.
13) Lines 287-289: maybe you can see differences between 0 and 25 oC using log scale.
Response: Thanks. I have tried the comparison using log scale and the curves between 0 and 25 ℃ becomes more closing.

14) In line 347, why a reference to fig. 7? In line 377, do you mean fig. 8 or fig. 9?

Response: Yes. This is indeed a reference to fig. 7.

Round 2

Reviewer 1 Report (New Reviewer)

I revised the manuscript “Molecular adsorption mechanism of water on the mineral surfaces of molybdenite (MoS2) based on Kinetic Monte-Carlo dynamics” for the second time. After the revision, the quality of the manuscript increased significantly, the authors did a great job to increase the clarity of the representation. They took my comments into account in part.

In my opinion, the comments which I gave to the paper in the first revision round are absolutely critical. Thus, I recommend a major revision for this manuscript and I hope that in the 3rd round the authors will fulfill all the requirements for a high-quality publication.

(1)    The reaction scheme which the authors employed for the kinetic Monte-Carlo simulations is still missing. The reaction scheme and the values of the reactions rate coefficients are important for the publication as they allow other scientists to reproduce the results of the current research. Thus I insist that the authors must include a table with elementary reaction (sorption/desorption/dissociation etc. ) and the values of rate coefficients which they used in the simulations. Such a table will facilitate the discussion in the manuscript as well as the authors can point to a particular reaction in the entry number N and show that this reaction is important/not important.

(2)    Equations. In equation (3), line 168 I again see “T” and “t”. In the answers to the reviewer the authors claim that “T” is the temperature, in the text they write “P is water vapor saturation pressure (KPa) and t is temperature (℃).”, line 169. This is very confusing and gives the impression that there is a mistake in the equation. I strongly suggest that the authors check all the equations, not only equation (3) and define all the symbols after the equation in the text of the manuscript.

(3)    Figure 10. The text on the figure, the captions of the axis are so small that I cannot read it. In general, the aspect ratio of the figures are wrong. Please correct this.

After the abovementioned comments are taken into account, the manuscript can be published.  

Author Response

Reviewer 1

Comments and Suggestions for Authors

I revised the manuscript “Molecular adsorption mechanism of water on the mineral surfaces of molybdenite (MoS2) based on Kinetic Monte-Carlo dynamics” for the second time. After the revision, the quality of the manuscript increased significantly, the authors did a great job to increase the clarity of the representation. They took my comments into account in part.

Response: Thank you very much for review.

In my opinion, the comments which I gave to the paper in the first revision round are absolutely critical. Thus, I recommend a major revision for this manuscript and I hope that in the 3rd round the authors will fulfill all the requirements for a high-quality publication.

  • The reaction scheme which the authors employed for the kinetic Monte-Carlo simulations is still missing. The reaction scheme and the values of the reactions rate coefficients are important for the publication as they allow other scientists to reproduce the results of the current research. Thus I insist that the authors must include a table with elementary reaction (sorption/desorption/dissociation etc. ) and the values of rate coefficients which they used in the simulations. Such a table will facilitate the discussion in the manuscript as well as the authors can point to a particular reaction in the entry number N and show that this reaction is important/not important.

Response: Thanks. We added a new table to include all the reaction scheme in KMC modeling and rate coefficient to help readers reproduce the result (Table 3). The original Table 3 containing full name of abbreviations was renamed as Table 4.

Table 3. Reaction scheme in kinetic Monte-Carlo dynamics and rate coefficient

Reaction

Rate coefficient (S-1)

MoS2 + H2O = MoS2-H2O

4.84 ×107

MoS2-H2O = MoS2 + H2O

3.62 × 1010

MoS2-H2O = MoS2-OH + H

2.8 × 1010

MoS2-OH + H = MoS2-H2O

2.9 × 1011

2H-MoS2 = MoS2 + H2

2.38 ×108

MoS2 + H2 = 2H-MoS2

1.20 ×108

  • In equation (3), line 168 I again see “T” and “t”. In the answers to the reviewer the authors claim that “T” is the temperature, in the text they write “P is water vapor saturation pressure (KPa) and t is temperature (℃).”, line 169. This is very confusing and gives the impression that there is a mistake in the equation. I strongly suggest that the authors check all the equations, not only equation (3) and define all the symbols after the equation in the text of the manuscript.

Response: Thanks for your advice. We checked all the equations. Many equations are represented in t and T, originally but we changed carefully based on T = t +273.15 in this manuscript.

(3)    Figure 10. The text on the figure, the captions of the axis are so small that I cannot read it. In general, the aspect ratio of the figures are wrong. Please correct this.

Response: Yes. The caption of axis of Fig. 10 are enlarged and the aspect ration are revised as you suggested.

After the above mentioned comments are taken into account, the manuscript can be published.  

Reviewer 3 Report (New Reviewer)

The revised version addresses some of my previous remarks, but the text is
still disorganized and written in poor English. Two examples are:
1) Lines 53 and 56 have two sentences beginning with "And".
2) In the response, the authors state that Figs. 1 and 2 are original results.
If so, they should not be presented in the Introduction, but in the Results
section.
This second example shows that the authors did not have a full understanding of
my previous comments on the manuscript organization: they moved comments on
Figs. 4-6 to the Results section, but did not realize that any other result
should also be moved to that section.
If the manuscript content is intended to be correctly understood by the readers,
extensive editing by a third part is necessary because the authors' have already
shown their inability to perform this task.

Regarding the scientific content, some points are still unclear:
1) How the results of Figs. 1 and 2 were obtained?
To make it clearer: In my previous report, I thought that Figs. 1 and 2 were
obtained by other authors, so I asked for references. In the response, the
authors argue that those figures are original results. Thus, the methods and
calculations leading to those results have to be shown.
2) The reason to use units mg/kg in Fig. 6 is unclear. The adsorbed amount in
mg/m^2 is the quantity that I expect to depend on pressure and temperature only.
However, different samples may have different specific surface areas, which
are measured in m2/kg. The plot in mg/kg is reasonable only if the isotherm
is calculated for a material with a well known specific surface area.
3) The authors' response regarding the use of log scale in Fig. 10 is confusing.
For instance, I cannot understand the meaning of "the curves ... become more closing".
The adsorption rate may vary some orders of magnitude, so its plot in log scale
may show the differences at the lowest temperatures that are hidden in Fig. 10.

Author Response

Reviewer 2

The revised version addresses some of my previous remarks, but the text is still disorganized and written in poor English. Two examples are:
1) Lines 53 and 56 have two sentences beginning with "And".

Response: Thanks. We removed the two “And” in the revised manuscript.

2) In the response, the authors state that Figs. 1 and 2 are original results. If so, they should not be presented in the Introduction, but in the Results section.

Response: Thanks. Fig. 1 and 2 are the Eh-pH diagrams calculated with the PHREEQC (Parkhurst and Appelo, 2013) and reaction forward modeling PHREEPLOT (Kinniburgh and Cooper, 2011). We added the method description about how to produce the two figures (section 2.1 and lines 89 to 93)

This second example shows that the authors did not have a full understanding of
my previous comments on the manuscript organization: they moved comments on
Figs. 4-6 to the Results section, but did not realize that any other result should also be moved to that section. If the manuscript content is intended to be correctly understood by the readers, extensive editing by a third part is necessary because the authors' have already
shown their inability to perform this task.

Response: Thanks. We also moved all the descriptions of Fig.1 and 2 into the result section in the revised manuscript (lines 184 to 196 in the revised manuscript).

Regarding the scientific content, some points are still unclear:
1) How the results of Figs. 1 and 2 were obtained? To make it clearer: In my previous report, I thought that Figs. 1 and 2 were obtained by other authors, so I asked for references. In the response, the authors argue that those figures are original results. Thus, the methods and
calculations leading to those results have to be shown.

Response: Thanks. Figs. 1 and 2 are indeed the original figures and the method how to produce them is also summarized in the section 2.1 in the revised manuscript (lines 89 to 93).

2) The reason to use units mg/kg in Fig. 6 is unclear. The adsorbed amount in mg/m^2 is the quantity that I expect to depend on pressure and temperature only. However, different samples may have different specific surface areas, which are measured in m2/kg. The plot in mg/kg is reasonable only if the isotherm is calculated for a material with a well known specific surface area.

Response: Yes. I am fully agreeing with this comment about the usage of unit about mg/kg or mg/m^2, especially for experimental study. But this study is based on the theoretical calculation with the assumption of ideal circumstance, when the surface area will not change with temperature. The reason using mg/kg is very convenient for readers then can easily estimate the adsorption amount based on the mass of adsorbent. And if they want to know the adsorption amount in mg/m^2, they can choose to measure the surface area if needed. If we present these data in mg/m^2, suggesting that surface area change as the function of temperature has considered for readers, actually which can’t be modelled at current stage.

3) The authors' response regarding the use of log scale in Fig. 10 is confusing.
For instance, I cannot understand the meaning of "the curves ... become more closing".
The adsorption rate may vary some orders of magnitude, so its plot in log scale
may show the differences at the lowest temperatures that are hidden in Fig. 10.

Response: Thank you very much for your advice. And we use the log scale in Fig. 10 a to clearly show the differences at the low temperature as you suggested . And two curves keep in parallel. This doesn’t change the previous conclusion, therefore we don’t change the text in the manuscript.

This manuscript is a resubmission of an earlier submission. The following is a list of the peer review reports and author responses from that submission.

Round 1

Reviewer 1 Report

I reviewed the previous version of this manuscript. In that review I listed several concerns regarding the quality of the theoretical modeling in the work. I don't believe that my prior concerns have been adequately addressed by the very minor revisions included in this version. Based on the author's response to my comments, I am not certain that they fully understand some of my concerns. I have two basic concerns that persist:

1)    The quality of the forcefield, specifically its ability to reproduce the molybdenite structure. The authors have only included DFT calculated unit cell parameters, not unit cell parameters computed with the UFF forcefield. This really needs to be done. The UFF forcefield was parameterized for Mo in a different valence state and you cannot assume the empirical potential is transferabable

2)    The quality of the forcefield, specifically its ability to reproduce ion-water interactions. Higher quality DFT calculations indicate substantial differences in adsorption energies from the present study. You cannot argue that a study that uses superior methods is incorrect, which is what this manuscript does. At the very least, this discrepancy needs to be acknowledged in the manuscript as a potential problem.   

In both instances the author’s response indicates that Jin et al., 2014 has done validated the UFF force field for molybdenite and molybdenite-water interactions. As I am unable to access this article at my institution, I cannot verify that this is the case, but based on what I can see I think it is unlikely.

For me there are two specific tasks that I feel need to be accomplished at the very least:

1)    Optimize the unit cell of molybdenite with the UFF forcefield and compare results to the reported DFT results.

2)    Describe efforts by Jin et al., and others to validate the force field model. Pay particular attention to the validation of the ion-water interactions, as the adsorption energies computed in this study are substantially more favorable than reported by Ghuman et al., 2015 using more accurate methods.

Reviewer 2 Report

The authors deal with the oxidation process of molybdenite (MoS2) under the weathering influence. Molecular dynamics and Monte-Carlo simulations were used to study the system. MoS2 and its oxidation process are exciting issues that may interest theoreticians and experimentalists. However, the manuscript has so many problems that I do not recommend it for publication. For example:

1.      The manuscript must be thoroughly reviewed. There are severe grammar and mistyping problems, and there are many phrases that are not clear or even not understandable.

2.      The authors poorly discuss the data and must carefully look at their results and try to describe them better.

3.      There are results wrongly presented and discussed.

These are some problems that I have noticed that make a promising work not suitable for publication.

Reviewer 3 Report

Review of minerals-1745469: Adsorption mechanism and kinetics of water molecule on the molybdenite (MoS2) and its implication on molybdenite weathering: an insight from the molecular dynamics simulation and kinetic Monte-Carlo

Comments:

·         As the authors mentioned in the Introduction section, the adsorption of water molecules on the molybdenite is very fast, for this reason, there is a lack of experimental reports on the adsorption process. Then, the results presented in the manuscript can be beneficial for understanding the adsorption mechanism. However, for any computational simulation study, the presented results must be supported by experimental data, at the least in an indirect form. In this sense, is it possible to carry out adsorption experiments, and the surface modification of molybdenite under normal pressure and temperature conditions can be followed by the XPS technique? These results could be correlated with the results shown in fig. 6, where is reported the water adsorbed in mg/kg.

·         Figures 1 and 2 can be omitted, with the references of such figures sufficient. On another side, if the figures are of their own authorship, details of their construction should be provided.

·         This reviewer supposes that the text in red in the manuscript refers to responses to another previous revision. In this sense, the last statement of the paragraph in red color on page 14 is out of context.

·         Is there any implication at the macroscopic level of the fact that the water molecules are adsorbed preferentially on the (110) surface?

·         Meaning of SIM?

·         In several parts of the manuscript, the chemical formula of water is spelled incorrectly (H2O). Revise subscripts.

        The correct form to mention products of the dissociation of water is H+ and OH. In several parts of the manuscript charge of the ions was omitted. Revise.
